# Effect of Printing Parameters on Mechanical Behaviour of PLA-Flax Printed Structures by Fused Deposition Modelling

**DOI:** 10.3390/ma14195883

**Published:** 2021-10-08

**Authors:** Yassine Elias Belarbi, Sofiane Guessasma, Sofiane Belhabib, Ferhat Benmahiddine, Ameur El Amine Hamami

**Affiliations:** 1UR1268 Biopolymères Interactions Assemblages, INRAE, F-44300 Nantes, France; yassine.belarbi@inrae.fr; 2GEPEA, UMR 6144, CNRS, Université de Nantes, F-44000 Nantes, France; sofiane.belhabib@univ-nantes.fr; 3LaSIE, UMR 7356 CNRS La Rochelle Université, CEDEX 01, 17042 La Rochelle, France; ferhat.benmahiddine1@univ-lr.fr (F.B.); ahamami@univ-lr.fr (A.E.A.H.); 44evLab, LaSIE, CNRS, EDF R&D La Rochelle University, CEDEX 01, 17042 La Rochelle, France

**Keywords:** fused filament deposition, PLA-flax, tensile behaviour, microstructure

## Abstract

Few studies have reported the performance of Polylactic acid (PLA) flax feedstock composite for additive manufacturing. In this work, we report a set of experiments conducted by fused filament technology on PLA and PLA-flax with the aim of drawing a clear picture of the potential of PLA-flax as a feedstock material. Nozzle and bed temperatures are both combined with the printing angle to investigate their influence on structural and mechanical properties. The study shows a low sensitivity of PLA-flax to process parameters compared to PLA. A varied balance between shearing and uniaxial deformation is found consistent with tensile results where filament crossing at −45/+45° provides the optimal load-bearing capabilities. However, Scanning Electron Microscopy (SEM) and high-speed camera recording shows a limiting reinforcing effect of flax fibre due to the presence of intra-filament porosity and a significant amount of fibre pull-out resulting from the tensile loading. These results suggest that the quality of the bond between PLA matrix and flax fibre, intra-filament porosity, and surface roughness should receive more attention as well as the need for more continuous fibre reinforcement in PLA filaments to optimise the performance of PLA-flax printed materials.

## 1. Introduction

The additive manufacturing technology is a process of joining materials layer-by-layer to achieve parts with a high degree of complexity compared to traditional processing routes such as extrusion and injection moulding [1,2]. Additive manufacturing or the commonly named technology 3D printing has already gained a mature state where several routes are now available to process a variety of materials such as stereolithography [3], selective laser melting [4], and fused filament deposition [5].

Among the feedstock materials that are widely considered for printing using fused filament deposition technology, one can name polylactic acid (PLA) [6] and acrylonitrile butadiene styrene (ABS) [7] as major feedstock materials. Polylactic acid (PLA) is one of the most used feedstock materials in fused filament technology. [8,9,10]. PLA is an affordable biosourced material obtained from starch. PLA is widely used in fused filament deposition because of its capacity for thermo-forming under relatively low thermal energy consumption. PLA also exhibits a fast crystallisation when cooled down, which makes it ideal for 3D printing. As a feedstock material, PLA has received significant attention since the early days of fused filament technology development. Some of the major areas of interest for PLA were the understanding of thermal kinetics (glass transition, crystallinity, melting) related to printing parameters [11,12] such as printing temperature and cooling rate [13,14]. Other contributions focused on the improvement of the mechanical performance of printed PLA [9,15]. In recent years, there was a steady tendency for blending PLA with different types of fillers to acquire a more reinforcing effect [16,17,18]. Among the systems that were developed, one should mention natural filler-based filaments such as a PLA-PHA blend with wood fibres and particles [17,19,20], PLA-jute and PLA-flax [21]. Most of the related development emphasis is on the critical point of filler-matrix compatibilisation to improve the performance and the short versus continuous reinforcement modes for fibre-based composites. Indeed, Hinchcliffe et al. [21] discussed the effect of flax reinforcing strands in 3D printed PLA prestressed composite. The authors showed the improvement of the specific mechanical performance of flax-based composites compared to jute-based composites. Le Duigou et al. [22] considered flax fibres as a reinforcing filler within PLA filament. The authors demonstrated the superior tensile performance (both stiffness and strength improvement) of continuous flax fibre-PLA composites with respect to PLA filament but highlighted the lack of transverse performance with respect to thermoprocessed composites. Zhang et al. [23] developed a customized setup to achieve continuous flax fibre reinforced thermoplastics composite. The authors reported substantial improvement of strength and stiffness both under tensile and flexural testing modes.

Early contributions in the field of additive manufacturing focused on the processing issues and particularly on finding optimal parameters to avoid significant loss in performance [7]. Some of the key findings were related to the anisotropy developed from varying the part orientation with regards to the filament organisation. For instance, a part subjected to tension in a particular direction should not encompass filament arrangement normal to this direction due the lack of cohesion along the material joints that triggers low tensile strength. Thus, inter inter-layer bond was a strong line of research and has led to numerous contributions aiming at understanding and improving the structuring of 3D printed materials [24]. Recent studies investigated the role of the microstructural organisation by means of 3D imaging techniques and showed the main characteristic of the defects in fused filament deposition [25]. This feature is the highly connected porous network as opposed to a low porosity content. If the filament arrangement remains a key factor in modulating the mechanical behaviour of printed structures, the intrinsic properties of the filaments constitute the other side of the equation.

Indeed, a recent study by the authors showed that the hemp-based filament can be regarded as a multi-phase system with a matrix, reinforcement, weak interface and internal porosity [11]. The combination of the weak interface and the internal porosity had a significant effect on the loss of performance. A similar result was obtained earlier by Le Duigou et al. [17] highlighting a limited mechanical performance especially with regards to tensile properties of printed 3D structures for which filaments were reinforced by wood fibres. This fact brought an additional motivation to consider other types of natural fibre reinforcement in 3D printing filaments. The main question to answer is: is the limiting reinforcement effect a common drawback of composite filaments with natural fibrous reinforcement? Starting from the idea that the specific performance of flax is better compared to hemp, which is mainly true with regards to the stiffness and strength [26], the aim of the present study is to determine if there is any limited mechanical transfer that can be associated with the use of flax fibres as a reinforcement material in PLA matrix.

## 2. Materials and Methods

The feedstock materials are commercially available PLA and flax-PLA filaments 1.75 mm in diameter, spool of 500 g and 1 kg for which the technical data sheet provided by the suppliers is summarised in Table 1. PLA was provided by Makershop company (Le Mans, France) under the tradename ECOFIL 3D and PLA-flax was provided by NANOVIA company (Louargat, France) for which the load content of flax is constant. The recommended printing temperatures for both filaments are in the ranges (195–215) °C and (210–260) °C, respectively. In addition, a bed temperature in the range (35–60) °C and (0–70) °C is recommended for PLA and PLA-flax, respectively. A printing speed of (30–60) mm/s is also part of the specifications for PLA-flax. Based on monitoring the weight loss from DSC/TGA experiments, the weight content of flax within the as-received filament is measured as 20%.

The printed geometry is a dogbone structure with 80 mm in length, 10 mm in gauge width and 4 mm in thickness (Figure 1).

The printing parameters for the dogbone structures are detailed in Table 2. Among the varied printing parameters, only the filament arrangement materialised by the printing angle (Θ) (Figure 1c), the printing (T_P_) and bed (T_B_) temperatures were considered. The building direction was set normal to the thickness of the specimen. The filament within the plane of construction (length–width) was arranged according to different layups corresponding to a printing angle Θ of 0°, 15°, 30° and 45°. The building sequence for Θ = 0° is +45°/−45° with respect to the length of the specimen. All printing conditions were combined to study possible cross effects. This means that a total of 24 conditions were tested for each feedstock material, which for the full factorial design adopted in this study concerns 48 runs. At least four samples per condition were printed, which means that a total of 128 specimens were tested for each feedstock material.

Two levels of bed temperatures were considered for both feedstock materials and three printing temperatures were also studied, and these are slightly above the supplier recommendation for PLA. These values represent a compromise because PLA-flax can be printed at higher temperatures compared to PLA. Using the same temperature range of PLA-flax for PLA is not recommended as a significant amount of stringing may occur.

Mechanical testing was performed on dogbone specimens using a uniaxial testing machine from Zwick Roell where a load cell of 10 kN is mounted (Figure 2). The loading rate was fixed to 5 mm/min for all conditions, and the testing was performed up to the rupture point. Young’s modulus, yield stress, tensile strength, elongation at break were the main mechanical quantities derived and related to the printing conditions. A sample observation during testing was planned in the setup where optical recording was performed using a high-speed camera (Phantom V7.3 from Photonline, Marly Le Roi, 78-France).

The purpose of these observations is double: study the deformation mechanisms under low frame rate and capture the mechanical instability at the rupture point under a configuration of high-speed recording. Thus, the frame rate varied between 100 and 42,000 fps, where fps refers to the number of frames per second. Only a partial view of the specimen was available at a high-speed rate, while the full frame (800 × 600 pixels) covering the entire specimen surface was easily achieved at lower rates.

The fractured samples were observed using an environmental Scanning Electron Microscope (SEM) to reveal the major microstructural alterations at the fractured surfaces and the filament arrangement characteristics at areas not affected much by the loading. The FEI quanta 200 ESEM/FEG Environmental microscope was used to complete this analysis. The preparation protocol did not require a metallisation stage. An acceleration voltage of 11kV and different magnification levels were chosen to observe the microstructure of the studied materials. Furthermore, pressures between 1.2 and 1.4 mbar were selected as reference parameters. The observations are performed normal to the fractured surfaces and with a tilt over the width of the specimens to study the filament arrangements near the fractured zones as well. SEM micrographs were acquired to quantitatively reveal the effect of printing angle, bed and printing temperatures.

## 3. Mechanical and Microstructural Results

The typical tensile performance of one replicate for both printed PLA and PLA-flax printed structures is depicted in Figure 3. The improvement of the tensile response ranking is evident more in the case of PLA (Figure 3a). The overall behaviour of printed PLA is elastic with a limited plasticity. It has to be mentioned that at the smallest printing angle, the rupture seems to be more delayed, which can be deduced by the progressive drop in tensile force. For PLA-flax (Figure 3b), it is more difficult to find a meaningful tendency as the overall effect of the printing angle is less evident and seems negligible. There is still proof of limited plasticity, but the ranking of all curves appears lower compared to the PLA case. This is the first proof of the limited reinforcing effect of flax and a low sensitivity to processing parameters compared to PLA. It has to be mentioned that the comparison between the two feedstock materials do not consider the possible differences in molecular weights where PLA grades in both filaments are not supposed to be the same. The limiting reinforcing effect of flax is rather considered from a microstructural viewpoint where interpretation should be bounded by the earlier mentioned consideration about PLA grades.

In addition to this main effect, a secondary effect on the ultimate properties is observed, which reveals a negative influence of Θ on both the elongation at break and the ultimate tensile stress for flax-PLA.

Figure 4 compares the snapshots of the deformed PLA-flax samples at different engineering strain levels for a combination of two printing angles and two bed temperatures. When the printing and bed temperatures are fixed to 200 and 50 °C (Figure 4a,b), differences between the rupture behaviours are observed. For Θ = 0°, the crack initiation from the external frame leads to a propagation with significant deviation from the opening mode. This deviation follows the filament arrangement pattern that, in this case, refers to a layup of −45°/45°. For Θ = 45°, crack instability follows a pure opening mode leading to a nearly flat ruptured profile. For both cases, near the rupture point, whitening of particular areas near the crack departure sites is observed. This change in the colour can be correlated to areas subjected to higher strain. For the case of PLA, it seems that these areas are wider compared to PLA-flax, concluding on a more diffuse damage.

Surprisingly, when the bed temperature is increased to 60 °C, the sample printed at Θ = 0° does not show signs of substantial deviation from the opening mode (Figure 4c). The lack of cohesion between filaments compared to the filament core should ease the cracking along the joints, which means that cracks running at 45° or −45° should to seen. Despite this disturbing fact, crack jaggedness seems to be more pronounced compared to the PLA-flax printed at Θ = 45° (Figure 4d).

In fact, further analysis of the cracking behaviour for this particular case is captured by high-speed recordings and confirms the minor tendency to follow the filament arrangement. The absence of substantial deviation can be interpreted in terms of the beneficial role of the compactness of the first layers brought by the improvement of the heating during the build. It is possible to measure the cracking speed between the two last frames for this condition (T_B_ = 60 °C), which seems to be rather slow compared to glassy materials. The measurement shows that the crack speed for this condition is about 240 m/s (Figure 5). At a lower bed temperature, the crack speed is doubled (550 m/s) even in the presence of crack deviation, which is supposed to delay the cracking progress.

Figure 6 shows the tensile response of a typical replicate from PLA and PLA-flax printed at various printing temperatures keeping the bed temperature constant. When the filament crossing is fixed to −45°/45° (i.e., Θ = 0°), both materials show opposite trends with respect to the printing temperature. Improvement of the inter-filament cohesion can explain the higher ranking of PLA tensile performance more obviously when the printing temperature is high. For PLA-flax, the increase in printing temperature induces the opposite effect. A rational explanation would call for an intrinsic modification at the filament scale, which is further discussed with the evidence of SEM micrographs. When the filament crossing produces a layup with 0°/90° (i.e., Θ = 45°), the ranking is still the same for both materials, which demonstrates consistency in the observations, but the overall ranking of the tensile responses is lower compared with Θ = 0°.

Further exploitation of the tensile results using SEM micrographs shows several distinctive features. For instance, the effect of the bed temperature on the deformed structures is captured in Figure 7. At the scale of the layup, signs of ductile rupture are observed based on the amount of stretching of the filaments (Figure 7a). From the framed window in Figure 7a, a closer view of the filament cross-section reveals multiple open-pits that are associated with the decohesion of the cross-linked filaments mostly by mechanisms of shearing (Figure 7b). Shearing takes place in this particular case due to the misalignment of the filaments with respect to the loading direction (Θ = 0°). At a larger bed temperature (T_B_ = 60°C), the amount of stretching is even higher (Figure 7c). This is confirmed by the positive correlation between the deformation at break and the bed temperature. In addition to these effects, there is no marked intensification of the pits population nor the increase in their dimensions (Figure 7d).

Figure 8 shows the same deformed configurations for the PLA-flax printed structures. The rupture profiles seem to be more jagged and no significant stretching can be depicted (Figure 8a). In addition, a closer view of the layup (Figure 8b) shows distinctive porous structures within the filaments. These porosities can be either a genuine defect of the as-received PLA-flax wire or develop during the printing process by a mechanism of fibre pull-out or also during the stretching of the filaments. A close view of the filament cross-section provides a clue about the role of the PLA-flax interface as a tentative explanation for the development of the inter-filament porosity. Indeed, voids surrounding the flax fibres indicate a lack of adhesion between the PLA matrix and the reinforcement. At a higher bed temperature (T_B_ = 60 °C), the rocky aspect of the ruptured surface also confirms a limited plasticity for PLA-flax printed structure (Figure 8d). A zoomed view on two particular areas shows the flax fibre configuration at the ruptured edge (Figure 8e,f). The overall diameter of the flax fibre ranges between 20 and 35 µm. Fibres of several hundreds of microns hanging on the surface can be seen, which means that flax fibres imbedded in PLA matrix exhibit a large aspect ratio (Figure 8e). Defibrillation can also be observed because of the partial pull-out (Figure 8f). There is still evidence of interfacial decohesion in most of the cases, indicating that the load transfer contributes both in deforming the raster and pulling out the fibres from their genuine environment.

Using a tilt option in SEM, the effect of the printing angle on the deformation of the raster is studied. Figure 9 shows the result achieved for all printing angles (Θ = 0°, 15°, 30°, 45°), using fixed printing and bed temperatures (T_P_ = 200 °C, T_B_ = 50 °C) for both PLA and PLA-flax. Figure 9a shows marked differences between the deformation of the raster and the external frame. According to the dogbone slicing procedure, an external frame consisting of two successive layers is combined with the raster. This frame provides mechanical stability to the entire structure. As can be seen in Figure 9a, the two layers of filaments are distinguishable. The transverse filament aspect ratio is consistent with the layer height–filament diameter ratio, which is kept at 0.5 for all conditions. In addition, the presence of inter-filament porosity within the frame is visible and refers to the overlapping generated by the elliptic forms of the filaments. The inter-filament porosity has an average size of the order of 100 µm. Unlike the raster tearing aspect, the frame exhibits a smooth rupture with limited stretching on the sides. This is because the deformation mode of the external frame is a uniaxial tension, whereas the raster is subject to a combination of shearing and tension. The same distinction between the raster and the external frame cannot be achieved for PLA-flax because of the overall limited stretching of the structure at the rupture point (Figure 9b). Even if it is barely seen from the perspective views in Figure 9b,d,f, the surface state of the flax-fibre filaments located at the periphery is rougher compared to the smooth aspect of the PLA filaments in Figure 9a. This would explain why the rupture patterns are more diffuse in PLA-flax. Indeed, crack growth would use the surface defects in PLA-flax fibres allowing the crack tip to continue through the filament instead of following the raster pattern. This helps unstable cracking under predominant mode I to develop at the expense of crack deviation under mixed mode. For a printing angle of 15°, the jaggedness of the ruptured surface is lower compared to an angle of 0° (Figure 9c). Although the same differences are observed between the PLA and PLA-flax samples (Figure 9c,d), there is a tendency towards lower tearing as the filament’s misalignment with respect to the loading direction decreases. This can be related to a certain extent to the decrease in the elongation at break observed from the tensile responses as the printing angle increases (Figure 3). The flatness of the ruptured surface becomes more evident for larger values of printing angles (Θ = 30°, Θ = 45°), as there are much more filaments that contribute to the load-bearing (Figure 9e–h). A continuity of the crack through the frame and the raster is clearly observed for Θ = 45° (Figure 9g,h).

Figure 10 compares the rupture patterns and filament arrangements of PLA and PLA-flax under high processing temperatures (T_P_ = 210 °C, T_B_ = 60 °C). A more cohesive structure is observed for PLA with a limited porosity between the raster and the external frame in the case of a small printing angle such as Θ = 0° (Figure 10a). The rupture pattern alternates rough and smooth surfaces indicating differences in deformation mechanisms following the same rational described above where tension is located at the external frame and inner filaments are associated with a flat rupture pattern. The cracking topography within the raster seems discontinuous and only tearing along the joints between the filaments prevails as to show signs of inter-filament decohesion. For PLA-flax under the same printing angle of 0° (Figure 10b), the rough fractured surface indicates a more continuous pattern for which no clear distinction can be foreseen between the cracking behaviour in the raster and the frame. No clear difference can be either observed between this case and the rupture patterns achieved under a lower printing temperature (Figure 10b). Figure 10c shows another view of the filament arrangement within the raster for PLA under the same high heating conditions and for a larger printing angle (Θ = 15°). Small porosity between the adjacent filaments of a typical size of 90 µm is observed because of the layup. A particular feature is the presence of a varied filament transverse morphology highlighted in the same figure. This varied morphology is related to the overlap in filament trajectory, which results in one filament deforming the adjacent one. This effect cannot be clearly observed in the case of PLA-flax under the same printing angle (Figure 10d), where the most highlighted effect remains the presence of a substantial amount of porosity within the filaments themselves even at a high printing temperature. Figure 10e shows the modification induced by a high processing temperatures and a large printing angle (Θ = 30°) on the PLA connectivity between the raster and the external frame. Porosities as large as 300 µm are observed, which feature the lack of material continuity between the frame and the raster. Smaller porosities of 100 µm also remain as a characteristic of the lack of cohesion between the filaments within the raster. For PLA-flax processed under the same conditions (Figure 10f), material discontinuity becomes substantial and takes the form of large gaps with no material connectivity along the filament length. In addition, load-bearing capability of the printed PLA-flax is limited to roughly half of the layups as only the filaments that are misaligned with 15° with respect to the loading direction contribute to the mechanical transfer. This situation is amplified for a printing angle of 45° (Figure 10g,h), where the load-bearing of transverse filaments relies on a limited contact area as can be observed for PLA (Figure 10g). The load-transfer is even more limited due to series of factors such as the intra-filament porosity, the material discontinuity within the raster and limited stretching of the filament for the case of PLA-flax (Figure 10h).

## 4. Discussion of Main Effects

The correlation between the mechanical properties and the printing parameters is studied through fitting procedure. Automated fitting routine based on both linear and nonlinear regressions performed considering the three printing parameters (T_B_, T_P_, Θ) showed that nonlinear complex equations with a large number of fitting parameters are top ranking ones. When comparing the number of parameters needed to the number of experiments available and the parameter levels used, it can be concluded that nonlinear regressions even representing the entire space of the parameters are not robust enough to provide a clear picture of the main printing parameter effects.

Thus, the main effects are analysed in this section with respect to first linear terms to achieve more clarity about the meaningful observed effects of PLA and PLA-flax feedstock materials. This means that parameter interactions from nonlinear terms are neglected in this analysis. Figure 11 shows the average measured mechanical quantities of both PLA and PLA-flax reported from the tensile testing experiments for all considered conditions. The average values for each mechanical property are obtained from four replicates of the considered printing parameter combinations. The analysis of scatter measured as the average of the standard deviation over the average value shows the following values 4%, 5%, 5%, 7%, and 5% for PLA Young’s modulus, yield stress, tensile strength, ultimate stress and elongation at break. This concludes the stable and reproducible results for PLA. The same is, in fact, achieved for PLA-flax where the scatter values are slightly higher: 1%, 3%, 3%, 14%, 7%.

The combined effect of Θ and T_B_ on Young′s modulus $E_{Y}$ of PLA is complex to understand for PLA. Indeed, at a lower T_B_, a positive correlation between $E_{\mathrm{PLA}}$ and Θ is achieved, whereas it tends to show the opposite for a higher T_B_. The combination of both effects leads to the following linear correlation, which can fairly be assumed to be not representing the true effect of bed temperature on the cohesiveness of the printed structure.
(1)$$E_{Y}\left(\mathrm{MPa}\right)=-0.16\times \Theta \left({}^{\circ}\right)-5.36\times T_{B}\left({}^{\circ}C\right)+1239\;R^{2}=0.65$$

All remaining mechanical quantities related to the behaviour of PLA tend to show a more beneficial effect of T_B_ and a negative correlation with Θ. It has to be mentioned that the trend exhibited by the elongation at break does not oppose the tendency observed for the tensile strength or the modulus of elasticity.
(2)$$\sigma_{Y}\left(\mathrm{MPa}\right)=-0.048\times \Theta \left({}^{\circ}\right)+0.49\times T_{B}\left({}^{\circ}C\right)+17\;R^{2}=0.91$$
(3)$$\sigma_{R}\left(\mathrm{MPa}\right)=-0.09\times \Theta \left({}^{\circ}\right)+0.53\times T_{B}\left({}^{\circ}C\right)+20\;R^{2}=0.90$$
(4)$$\varepsilon_{R}\;\left(\%\right)=-0.04\times \theta \left({}^{\circ}\right)+0.53\times T_{B}\left({}^{\circ}C\right)-0.43\;R^{2}=0.88$$
where $\sigma_{Y}$, $\sigma_{R}$, and $\varepsilon_{R}$ are yield stress, tensile strength and elongation at break.

The correlations achieved for the printing temperature show also an improvement of the PLA stiffness, yielding, strength and a decrease in the elongation at break when the printing temperature increases. This correlation is combined with an overall negative effect between these quantities and the printing angle. These correlations also have a limited validity within the following bounds of processing parameters, namely T_P_ (200–210) °C and Θ (0–45).
(5)$$E_{Y}\left(\mathrm{MPa}\right)=-0.56\times \Theta \left({}^{\circ}\right)+10.42\times T_{P}\left({}^{\circ}C\right)-1144\;R^{2}=0.89$$
(6)$$\sigma_{Y}\left(\mathrm{MPa}\right)=-0.04\times \Theta \left({}^{\circ}\right)+0.11\times T_{P}\left({}^{\circ}C\right)+22.9\;R^{2}=0.72$$
(7)$$\sigma_{R}\left(\mathrm{MPa}\right)=-0.03\times \Theta \left({}^{\circ}\right)+0.13\times T_{P}\left({}^{\circ}C\right)+24.4\;R^{2}=0.76$$
(8)$$\varepsilon_{R}\;\left(\%\right)=-0.02\times \Theta \left({}^{\circ}\right)-0.99\times T_{P}\left({}^{\circ}C\right)+26.9\;R^{2}=0.76$$

It can be concluded at this stage that for PLA, the correlation between the tensile properties and the printing parameters can be justified by the improvement of the cohesive structure and the raster configuration that both modulate the shearing and tension according to the filament misalignment. The qualitative interpretation from SEM micrographs is supported by the linear relationships determined for each mechanical quantity. For PLA-flax, the parametric chart in Figure 11b shows a rather contrasted situation for which only the printing parameter combinations on the left would result in an improvement of the tensile performance. This means, for instance, that T_P_ = 200 °C seems to be an optimal condition for this material. However, Figure 11 shows that even with this optimal condition, PLA-flax does not bring a larger mechanical benefit compared to pure PLA since all mechanical properties are below those of PLA. The benefit would be towards thermal properties, which are beyond the scope of the present study.

If the derived correlations between T_B_, Θ and the tensile properties are considered, these would show a limited variation with respect to the same correlations established for PLA. They would also show linear effects with much less confidence. In addition, the same contrasted effect of T_B_ on stiffness is observed and this one tends to promote larger Θ combined with higher Θ. For the remaining mechanical quantities, a decrease in Θ results in an overall improvement of the PLA-flax tensile performance.
(9)$$E_{Y}\left(\mathrm{MPa}\right)=0.21\times \Theta \left({}^{\circ}\right)+2.65\times T_{B}\left({}^{\circ}C\right)+712\;R^{2}=0.59$$
(10)$$\sigma_{Y}\left(\mathrm{MPa}\right)=-0.04\Theta \times \left({}^{\circ}\right)+0.27\times T_{B}\left({}^{\circ}C\right)\;23.3\;R^{2}=0.75$$
(11)$$\sigma_{R}\left(\mathrm{MPa}\right)=-0.07\Theta \times \left({}^{\circ}\right)+0.22\times T_{B}\left({}^{\circ}C\right)+31\;R^{2}=0.73$$
(12)$$\varepsilon_{R}\;\left(\%\right)=-0.03\Theta \times \left({}^{\circ}\right)-0.07\times T_{B}\left({}^{\circ}C\right)+3.4\;R^{2}=0.93$$

The achieved correlations for T_P_ are slightly different from the PLA case. While an overall positive effect of the filament crossing at +45°/−45° would be the most appropriate choice (i.e., small Θ), the increase in T_P_ is found to be an issue for most measured mechanical quantities. This would contrast with the idea of improving the cohesion between the filament by higher thermal input from the process. As shown in the review work by Gao et al. [24], the quality of the bond and bonding formation mechanism varies from one material to another, which results in sparse effects on mechanical anisotropy. However, since the printing concerns a composite material with some amount of flax fibre, the SEM observations tend to suggest that fibre pull-out and intra-filament porosity development may be substantially increased by a higher temperature. This would be a rational explanation to a negative effect of T_P_ under the circumstances that the quality of the PLA-flax interface remains weak at higher T_P_.
(13)$$E_{Y}\left(\mathrm{MPa}\right)\;=\;0.40\times \Theta \left({}^{\circ}\right)-2.61\times T_{P}\left({}^{\circ}C\right)+1378\;R^{2}=0.51$$
(14)$$\sigma_{Y}\left(\mathrm{MPa}\right)\;=-0.04\times \Theta \left({}^{\circ}\right)-0.05\times T_{P}\left({}^{\circ}C\right)+49\;R^{2}=0.34$$
(15)$$\sigma_{R}\left(\mathrm{MPa}\right)\;=-0.09\times \Theta \left({}^{\circ}\right)-0.08\times T_{P}\left({}^{\circ}C\right)+60\;R^{2}=0.61$$
(16)$$\varepsilon_{R}\;\left(\%\right)\;=-0.04\times \Theta \left({}^{\circ}\right)-0.02\times T_{P}\left({}^{\circ}C\right)+12\;R^{2}=0.70$$

Further exploration of the genesis of the intrafilament porosity is performed by optical imaging of the as-received PLA-flax filament shows that the presence of globular porosity of a typical size of 85 µm goes back to the fabrication process of the as-received filament. Indeed, the transverse cut in the filament’s prior extrusion (Figure 12) highlights this characteristic feature of the PLA-flax filament, which leaves only the small voids to be generated from the fibre pulling out.

## 5. Conclusions

This study confirms that the reinforcing effect of flax induces a series of issues related to the presence of intra-filament porosity, significant gaps, surface state and limited stretching of the filament. Using the leverage of printing and bed temperature only results in limited benefit compared to PLA, where tensile properties are found less sensitive than expected. The most contributing factor is the filament arrangement materialised by the printing angle. Both types of filaments PLA and PLA-flax do share the same overall effect, where a layup of −45°/+45° provides the optimal load-bearing capability as all filaments participate in the load transfer. In fact, this load transfer can be foreseen as a combination of shearing and tension for which the shearing contribution tends to vanish when the printing angle is increased to 45° resulting in half of the filaments completely misaligned with respect to the loading direction. Printability of PLA-flax under high printing temperature seems to be an issue, which may be explained by the amount of fibre pull-out and the intra-filament porosity observed by SEM. All these downsides of processing would direct future research into ways to improve the filament roughness, reduce the intra-filament porosity and improve the reinforcing effect by privileging filaments with continuous fibres. With regards to the last point, even though continuous fibres are promising from the standpoint of intrinsic performance, these still represent an open research field as their use in filament may result in reduced processability and more structural defects. Surface modification or pretreatment should also receive increased attention as an effective way of improving the performance of natural fibre-based 3D printed composites.

## Figures and Tables

**Figure 1 materials-14-05883-f001:**
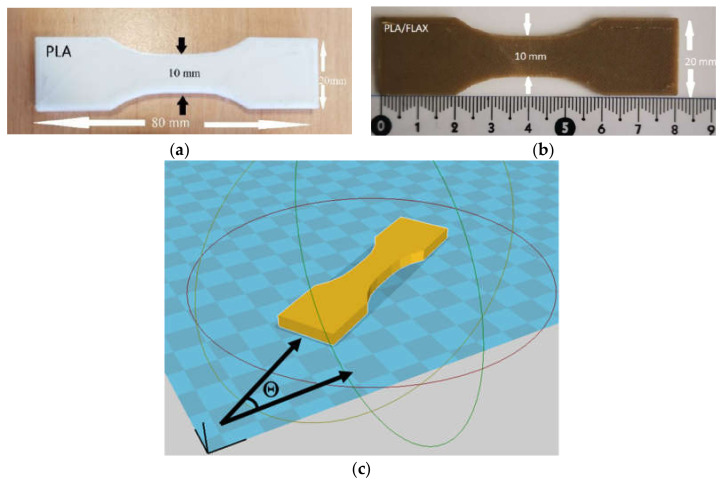
Dogbone geometry and related dimensions. (**a**) PLA, (**b**) PLA-flax feedstock materials, (**c**) definition of printing angle Θ.

**Figure 2 materials-14-05883-f002:**
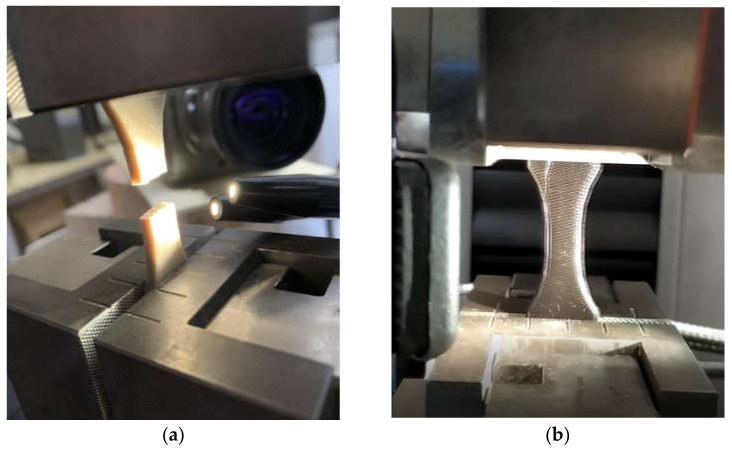
Experimental setup showing the tensile testing configuration with an optical camera and arc lamp light source for high-speed recording: (**a**) PLA-flax ruptured sample; (**b**) Illuminated PLA prior testing showing the filament orientation.

**Figure 3 materials-14-05883-f003:**
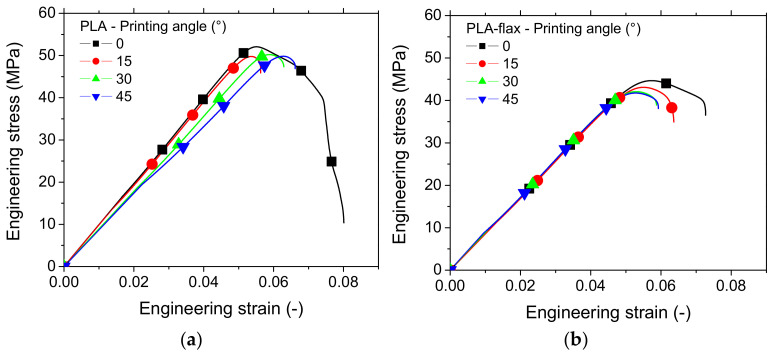
Comparison between the tensile performance of PLA and PLA-flax samples printed with the following conditions: T_P_ = 200 °C, T_B_ = 60 °C at various printing angles: (**a**) PLA; (**b**) PLA-flax.

**Figure 4 materials-14-05883-f004:**
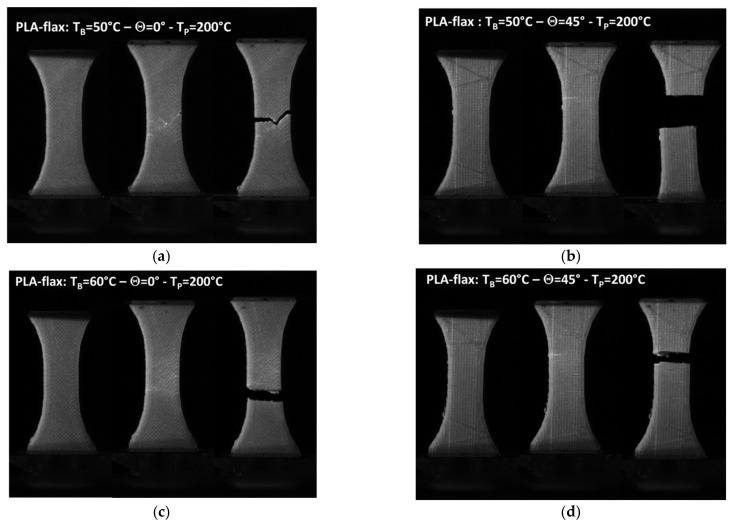
Snapshots of the deformed PLA-flax captured at 100 fps at different load levels (prior loading, prior rupture point, at rupture point) for T_P_ = 200 °C, and various combinations of bed temperature and printing angle: (**a**) T_B_ = 50 °C-Θ = 0°; (**b**) T_B_ = 50 °C-Θ = 45°; (**c**) T_B_ = 60 °C-Θ = 0°; (**d**) T_B_ = 60 °C-Θ = 45°.

**Figure 5 materials-14-05883-f005:**
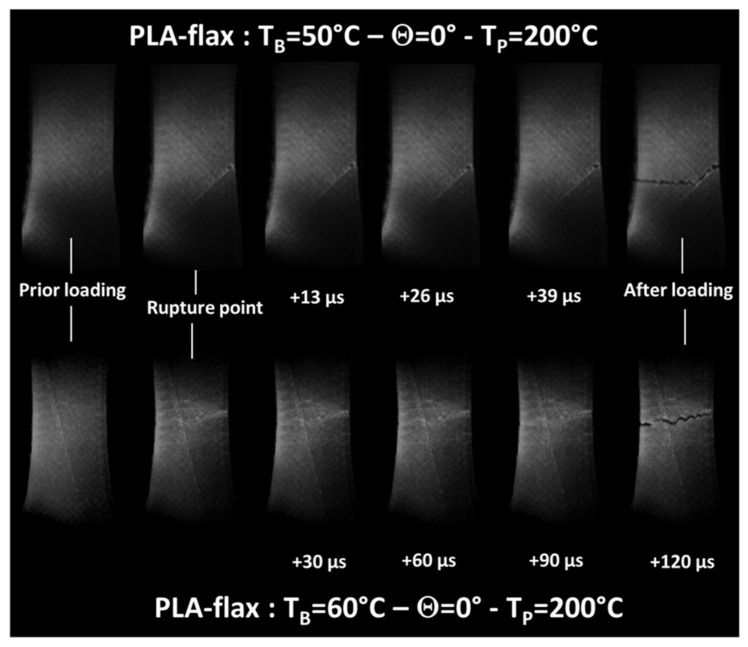
Cracking behaviour captured at 33k–42k fps of the deformed PLA-flax for two bed temperatures (Θ = 0°, T_P_ = 200 °C).

**Figure 6 materials-14-05883-f006:**
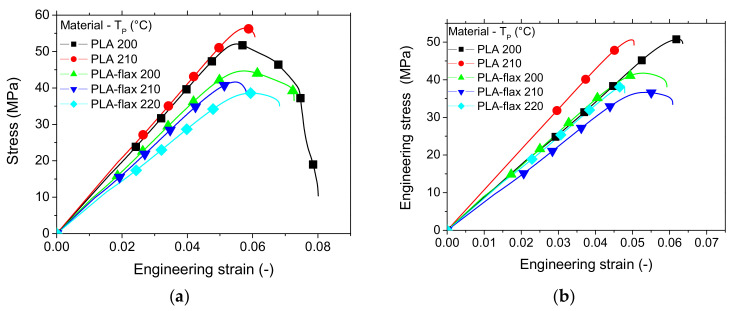
Effect of printing temperature on the tensile behaviour of PLA and PLA-flax printed materials (T_B_ = 60 °C) for two different printing angles: (**a**) Θ = 0°; (**b**) Θ = 45°.

**Figure 7 materials-14-05883-f007:**
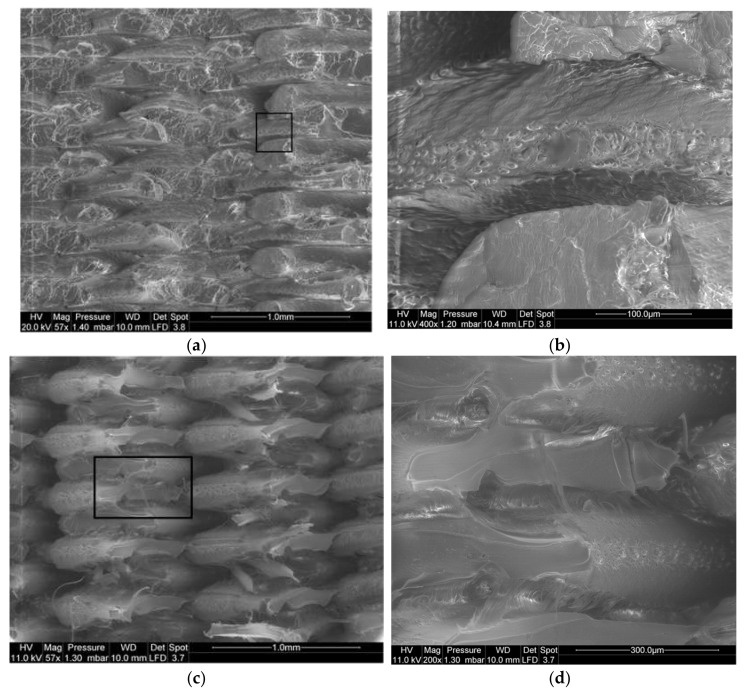
SEM micrographs showing the effect of bed temperature on the topography of PLA fractured surfaces (T_C_ = 200 °C, Θ = 0°): (**a**) T_B_ = 50 °C scale of filament layup; (**b**) T_B_ = 50 °C zoomed-view on filament cross-section; (**c**) T_B_ = 60 °C scale of filament layup; (**d**) T_B_ = 60 °C zoomed-view on filament cross-section.

**Figure 8 materials-14-05883-f008:**
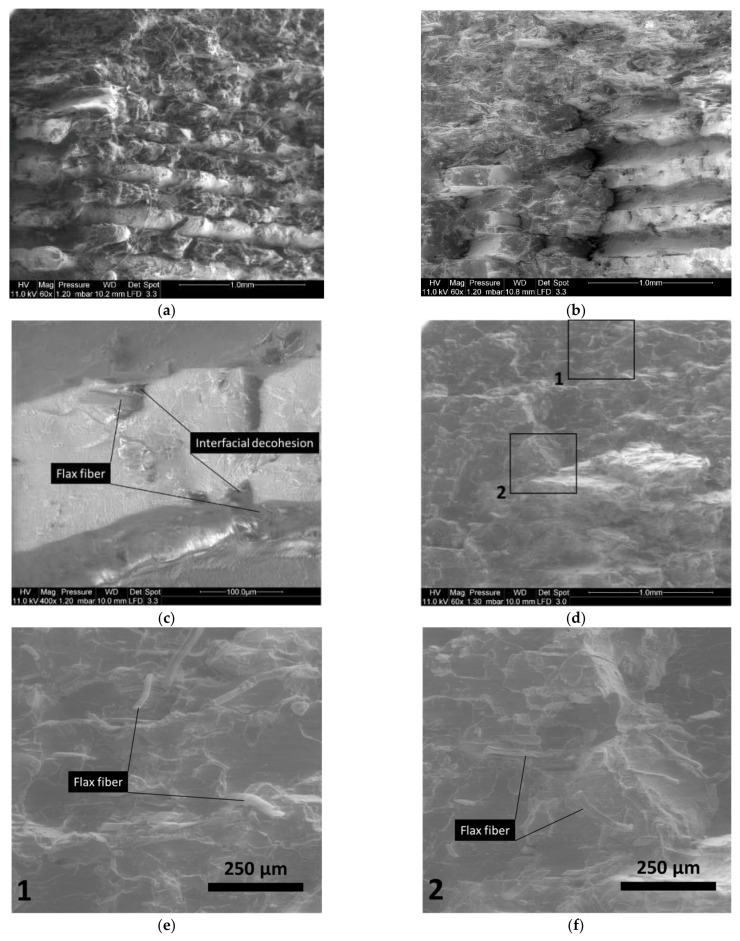
SEM micrographs showing the effect of bed temperature on the topography of PLA-flax fractured surfaces (T_C_ = 200 °C, Θ = 0°): (**a**) T_B_ = 50 °C scale of several filament layup; (**b**) T_B_ = 50 °C a closer view of the filament layup; (**c**) T_B_ = 50 °C zoomed-view on filament cross-section; (**d**) T_B_ = 60 °C scale of filament layup with two framed areas; (**e**) T_B_ = 60 °C zoomed-view of the flax fibre structure within the raster; (**f**) T_B_ = 60 °C another location on the same micrograph showing flax fibres.

**Figure 9 materials-14-05883-f009:**
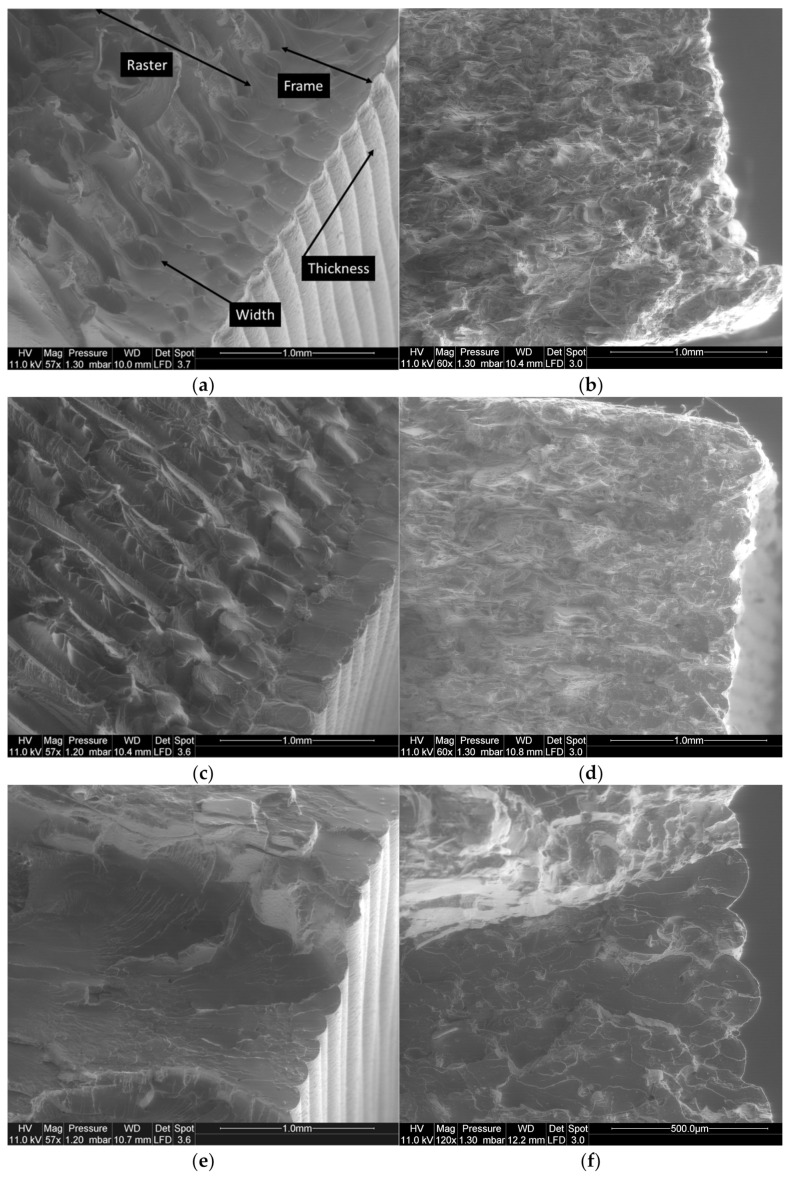
SEM micrograph showing the effect of printing angle on the topography of PLA and PLA-flax fractured surfaces (T_P_ = 200 °C, T_B_ = 60 °C): (**a**) PLA Θ = 0°; (**b**) PLA-flax Θ = 0°; (**c**) PLA Θ = 15°; (**d**) PLA-flax Θ = 15°; (**e**) PLA Θ = 30°; (**f**) PLA-flax Θ = 30°; (**g**) PLA Θ = 45°; (**h**) PLA-flax Θ = 45°.

**Figure 10 materials-14-05883-f010:**
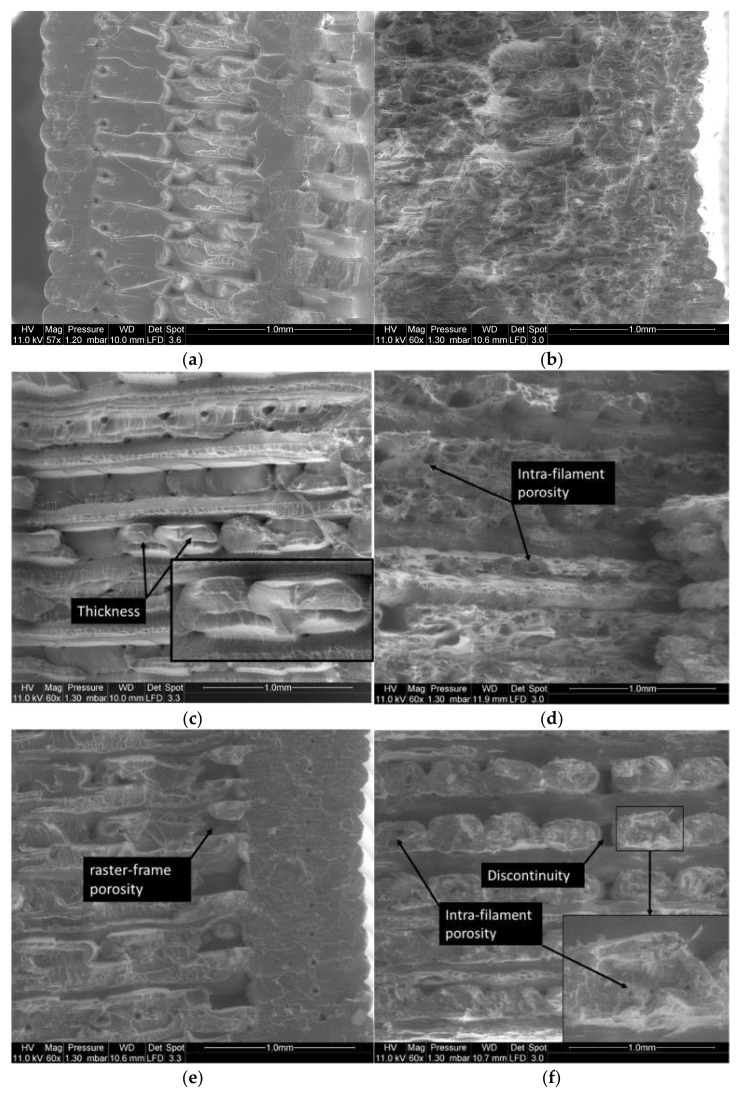
SEM micrographs showing the effect of high printing and bed temperatures on the topography of PLA and PLA-flax fractured surfaces (T_P_ = 210 °C, T_B_ = 60 °C): (**a**) PLA Θ = 0°; (**b**) PLA-flax Θ = 0°; (**c**) PLA Θ = 15°; (**d**) PLA-flax Θ = 15°; (**e**) PLA ν = 30°; (**f**) PLA-flax Θ = 30°; (**g**) PLA Θ = 45°; (**h**) PLA-flax Θ = 45°.

**Figure 11 materials-14-05883-f011:**
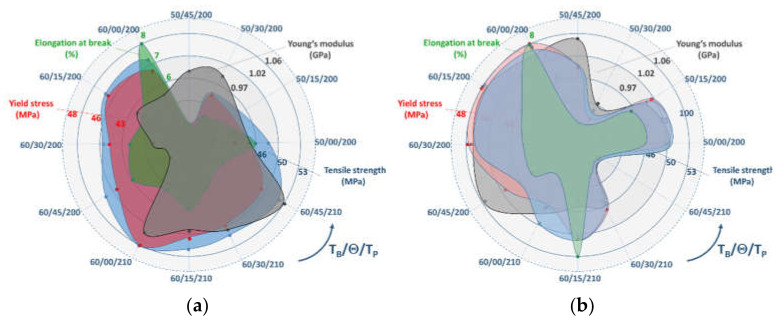
Parametric chart showing the dependence of the tensile properties on printing conditions for: (**a**) PLA; (**b**) PLA-flax.

**Figure 12 materials-14-05883-f012:**
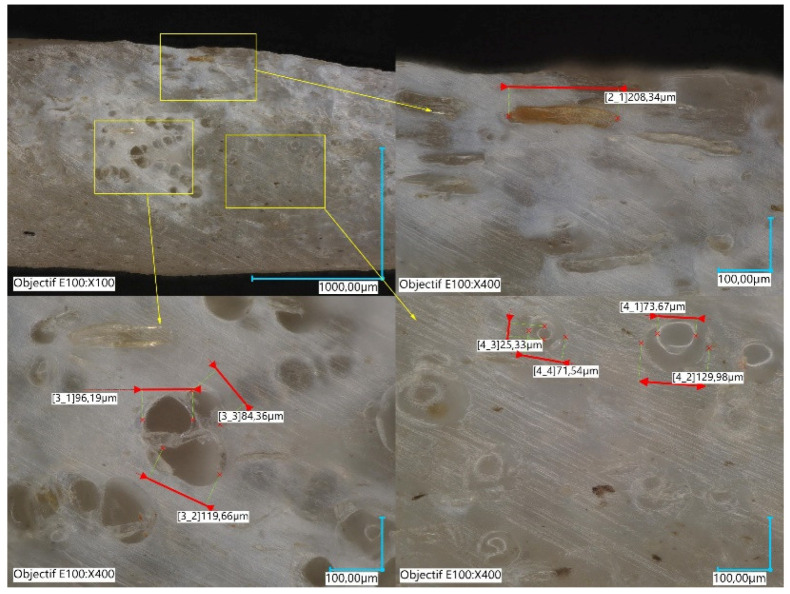
Intra-filament voids in the as-received PLA-flax filament observed using the Keyence VHX 7000 microscope.

**Table 1 materials-14-05883-t001:** Main characteristics of studied feedstock materials according to the supplier’s data sheet.

Property	PLA	PLA-Flax
Density, (g/cm^3^)	1.24	1.07
Moisture absorption (ppm)	1968	-
Melting temperature (°C)	115 ± 35	-
Glass transition (°C)	57	54
MFR ^1^ (gr/10 min)	9.56	-
Tensile modulus, (MPa)	3384	3400
Tensile strength, (MPa)	68	
Flexural modulus, (MPa)	-	2300
Flexural strength (MPa)	-	39
Impact strength ^2^, (3.4 kJ/m^2^)	-	-
Elongation at break, (%)	3	2
Hardness, (Shore D)	-	77

^1^ melt flow rate at 210 °C/2.16 kg. ^2^ Sharpy method 23 °C.

**Table 2 materials-14-05883-t002:** Printing conditions considered for PLA and PLA-flax.

Parameter	Value	Parameter	Value
Layer height	0.2 mm	Filament diameter	1.75 mm
Wall thickness	0.8 mm	Flow	100%
Top/Bottom thickness	0.6 mm	Nozzle diameter	0.4 mm
Infill density	100%	Build plate addition	None
Print speed	50 mm/s	Printing orientation	Modified on Cura
Travel speed	90 mm/s	Bed temperature	50 °C or 60 °C
Support	Disable	Printing temperature	200 °C, 210 °C, 220 °C

## Data Availability

Raw data used to produce the results of the study are available from the authors upon request.

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
