# Peer review of "Effect of Printing Parameters on Mechanical Behaviour of PLA-Flax Printed Structures by Fused Deposition Modelling"

_materials, 2021, doi:10.3390/ma14195883_

Round 1
Reviewer 1 Report
Belarbi et al. report the effect of printing parameters on the mechanical behavior of PLA-flax composites by FDM printing. The printing parameters of nozzle and bed temperatures are combined with printing angle to investigate their influence on structural and mechanical properties. The quality of the bond between PLA matrix and flax fibre, intra-filament porosity, and surface roughness are identified to be associated with the limited mechanical properties of the composites similar to many other systems. The experiments were systematically conducted and the writing is clear. I recommend accepting the manuscript after addressing the following issues:
- The current introduction is too brief. A bit of background in using PLA and natural fiber reinforced in FDM printing can be added. In addition, the progress in 3D printing of natural fiber-reinforced composite, particularly, the state of art progress in 3d printing continuous flax/PLA biocomposite should be included.
- What is the fiber loading and length of flax fiber in the PLA-flax? The presence of fiber influences printing and morphology substantially. Is it possible to use PLA-flax feedstock from the same brand with two different fiber loading to study the effect of fiber on the mechanical properties? Also, the direct comparison between PLA and PLA-flax can be a problem if the more specific material information, such as molecular weight is missing.
- Is the intra-filament porosity from the material itself or the printing process? Some of the marked intra-filament pores are large (Figure 10d) and some are small (Figure 10f). The small one looks like the hole by fiber pulling out. The SEM images of the original PLA-flax feedstock can be provided to confirm if some pores exist in the original feedstock.
- The author mentioned that the quality of the bond between the matrix and fibre is important. Some discussion on improving the interface can be added. Relevant reference(https://www.sciencedirect.com/science/article/abs/pii/S2214860420310307) can be helpful in such a discussion. The use of longer fiber may result in reduced processability and more structural defects, which can be found in the literature. Instead, surface modification or pretreatment may be more effective.
Author Response
Reviewer 1:
Belarbi et al. report the effect of printing parameters on the mechanical behavior of PLA-flax composites by FDM printing. The printing parameters of nozzle and bed temperatures are combined with printing angle to investigate their influence on structural and mechanical properties. The quality of the bond between PLA matrix and flax fibre, intra-filament porosity, and surface roughness are identified to be associated with the limited mechanical properties of the composites similar to many other systems. The experiments were systematically conducted and the writing is clear. I recommend accepting the manuscript after addressing the following issues:
We would like to thank the reviewer for his positive consideration of our manuscript. He will find below our answer regarding his comments.
The current introduction is too brief. A bit of background in using PLA and natural fiber reinforced in FDM printing can be added.
Done, thank you for the comment, we were able to improve the discussion about PLA and natural fibre reinforcement with the help of supporting literature work.
Amendment :
Introduction section : “Polylactic acid (PLA) is one of the… compatibilization to improve the performance”
Added references
- Le Duigou, M. Castro, R. Bevan, N. Martin, 3D printing of wood fibre biocomposites: From mechanical to actuation functionality, Materials & Design, 96 (2016) 106-114.
- Milosevic, D. Stoof, K. Pickering, Characterizing the Mechanical Properties of Fused Deposition Modelling Natural Fiber Recycled Polypropylene Composites, Journal of Composites Science, 1 (2017) 7.
- A. Hinchcliffe, K.M. Hess, W.V. Srubar, Experimental and theoretical investigation of prestressed natural fiber-reinforced polylactic acid (PLA) composite materials, Composites Part B: Engineering, 95 (2016) 346-354.
- Kariz M, Sernek M, Obućina M, and Kuzman MK. Materials Today Communications 2018;14:135-140.
- Mazzanti V, Malagutti L, and Mollica F. Polymers 2019;11(7):1094.
- Guessasma; Belhabib; Nouri, Microstructure and Mechanical Performance of 3D Printed Wood-PLA/PHA Using Fused Deposition Modelling: Effect of Printing Temperature. Polymers 2019, 11, (11), 1778.
In addition, the progress in 3D printing of natural fiber-reinforced composite, particularly, the state of art progress in 3d printing continuous flax/PLA biocomposite should be included.
We do agree with the reviewer remark, we added some specific discussion about Pla-flax feedstock material starting by the outputs from the following references
Hinchcliffe, S. A.; Hess, K. M.; Srubar, W. V., Experimental and theoretical investigation of prestressed natural fiber-reinforced polylactic acid (PLA) composite materials. Composites Part B: Engineering 2016, 95, 346-354.
Le Duigou, A.; Barbé, A.; Guillou, E.; Castro, M., 3D printing of continuous flax fibre reinforced biocomposites for structural applications. Materials & Design 2019, 180, 107884.
Zhang, H.; Liu, D.; Huang, T.; Hu, Q.; Lammer, H., Three-Dimensional Printing of Continuous Flax Fiber-Reinforced Thermoplastic Composites by Five-Axis Machine. Materials 2020, 13, (7), 1678.
Amendment in introduction section: “Hinchcliffe et al …flexural testing modes.”
What is the fiber loading and length of flax fiber in the PLA-flax?
Based on monitoring the weight loss from DSC/TGA experiments, the weight content of flax within the as-received filament is measured as 20%.
Amendment : section 2 : “Based on monitoring the weight…20%.”.
The presence of fiber influences printing and morphology substantially. Is it possible to use PLA-flax feedstock from the same brand with two different fiber loading to study the effect of fiber on the mechanical properties?
This is a good idea. However, the authors considered a commercially available filament where the weight content of flax fibre is constant.
Amendment : section 2: “for which the load content of flax is constant.”
Also, the direct comparison between PLA and PLA-flax can be a problem if the more specific material information, such as molecular weight is missing.
This is true, especially if PLA grades for both materials were not the same. As we mentioned in the introduction section the motivation was to study the limiting reinforcement of natural fibres that are commercially available in the market. We know that such comparison will not be enough accurate without considering the same matrices as a common ground. However, as the reviewer may notice the gap noticed between the two feedstock materials is more associated to the presence of interfacial porosities and voids within the PLA-flax filament and we believe that this is a driving factor for the lowering of the PLA-flax performance.
Amendment in section 3: “It has to be mentioned that the comparison between the two feedstock materials do not consider the possible differences in molecular weights where PLA grades in both filaments are not supposed to be the same. The limiting reinforcing effect of flax is rather considered from a microstructural viewpoint where interpretation should be bounded by the earlier mentioned consideration about PLA grades.”
Is the intra-filament porosity from the material itself or the printing process? Some of the marked intra-filament pores are large (Figure 10d) and some are small (Figure 10f). The small one looks like the hole by fiber pulling out. The SEM images of the original PLA-flax feedstock can be provided to confirm if some pores exist in the original feedstock.
We considered further optimal images to further investigate the genesis of the intrafilament porosity. The results show that the presence of large globular voids goes back to the as-received filament as demonstrated from the following transverse cut in the filament prior extrusion
Figure 1. Intra-filament voids in the as-received PLA-flax filament observed using Keyence VHX 7000 microscope.
Amendment in section 3: “Further exploration of the… fibre pulling out.” + Figure 12 added.
The author mentioned that the quality of the bond between the matrix and fibre is important. Some discussion on improving the interface can be added. Relevant reference(https://www.sciencedirect.com/science/article/abs/pii/S2214860420310307) can be helpful in such a discussion.
We thank the reviewer for the additional reference which was added to the discussion of main effects.
Amendment reference citation in introduction section : “Thus, inter inter-layer bond was…printed mateials [24]”. + section3 “As shown in the review work… effects on mechanical anisotropy”.
The use of longer fiber may result in reduced processability and more structural defects, which can be found in the literature. Instead, surface modification or pretreatment may be more effective.
We do agree with this analysis and we share the viewpoint of the reviewer.
Amendment in Conclusion section: “With regards to the last point, continuous… fibre-based 3D printed composites.”

Reviewer 2 Report
This experimental study shows how the mechanical properties of PLA-flax printed parts are affected by a set of selected parameters. The combination of tensile tests, crack evolution analysis using a high-speed camera, and fracture inspection through microphotographs provide enough experimental material to claim that filament angle deposition is the most influential parameter. Moreover, the PLA-flax reinforcement seems insufficient to improve parts printed at high hot-end and bed temperature, unlike in regular PLA is observed, due to fiber pull-out and intra-filament porosity noted by the authors.
The paper helps to understand the relation among microstructure, printing parameters, and mechanical properties of laminated printed parts. The manuscript is well organized, with graphics to clarify the experimental results and a detailed discussion (too long, in my opinion). There are also some issues, which corrected, could improve the paper quality:
- Section 2, page 3, lines 87-89. The design of experiments is not clear. According to the factors (filament, hot-end temperature, bed temperature, and angle) and levels (2, 3, 2, and 4 respectively), for a full factorial design, we have 48 runs, with 4 replications 192 samples. On the other hand, the authors chose this design to identify cross effects, but this has not been developed in the manuscript. Finally, if there are replications, are the values portrayed in figures 3, 6, and 11 mean values? Furthermore, which deviations were observed by the authors?
- Page 3, line 93. Why have the authors chosen printing temperatures "slightly above" the manufacturer's recommendations? And Why those specific values?
- Page 3, Table. It is Table 2. Layer height is included two times in the table. Finally, a graphical scheme about the filament deposition angle and the building sequence could help identify those parameters with part orientation in the printer bed and the angle for the linear infill pattern used.
- Section 3. Regarding the regression equations, did the authors try to accomplish a regression for the three parameters: angle, Tp, and Tb, at the same time?
- Section 4, lines 378-379. In the experiments, did the authors identify any mechanical benefit of using flax? If yes, this is not clear in the previous sections.
- Figure 4. Could the authors be able to substitute the black background with the white color?
- Figure 7.d, Figure 8. b Which regions are zoomed?
- Figure 11. Figures labels should be (a) and (b).
Author Response
Reviewer 2:
This experimental study shows how the mechanical properties of PLA-flax printed parts are affected by a set of selected parameters. The combination of tensile tests, crack evolution analysis using a high-speed camera, and fracture inspection through microphotographs provide enough experimental material to claim that filament angle deposition is the most influential parameter. Moreover, the PLA-flax reinforcement seems insufficient to improve parts printed at high hot-end and bed temperature, unlike in regular PLA is observed, due to fiber pull-out and intra-filament porosity noted by the authors.
The paper helps to understand the relation among microstructure, printing parameters, and mechanical properties of laminated printed parts. The manuscript is well organized, with graphics to clarify the experimental results and a detailed discussion (too long, in my opinion). There are also some issues, which corrected, could improve the paper quality:
We thank the reviewer for his opinion. We are sorry for the length of discussion which is mainly due to the interdependency of the parameters we have selected (printing temperature, bed temperature, printing angles) and for the comparison between two feedstock materials PLA and flax-PLA. We tried to organize the manuscript to allow a better reading of the main finding by separating the results from the discussion. We also wanted to provide evidence for each statement we have found to help the readers to understand the main effects. If there are particular sections where we can be more concise we would be happy to consider further improvement. Here you will find our detailed modification based on the reviewer remarks.
- Section 2, page 3, lines 87-89. The design of experiments is not clear. According to the factors (filament, hot-end temperature, bed temperature, and angle) and levels (2, 3, 2, and 4 respectively), for a full factorial design, we have 48 runs, with 4 replications 192 samples.
We are sorry for this confusion we meant per feedstock material. The reviewer is right the total number of combination is 48.
Amendment in section 2 : “This means that a total of 24 conditions were tested for each feedstock material, which for the full factorial design adopted in this study concerns 48 runs. At least four samples per condition were printed, which means that a total of 128 specimens were tested for each feedstock material.”
On the other hand, the authors chose this design to identify cross effects, but this has not been developed in the manuscript.
It has been shown in multiple places. For instance in section 3 we studied the combined effect of prining andle and bed temperature
Former version :
Cross effect of printing angle and bed temperature: Figure 4 compares the snapshots of the deformed PLA-flax samples at different engineering strain levels for a combination of two printing angles and two bed temperatures.”
Cross effect of printing temperature and printing angle: Figure 6 shows the tensile response of PLA and PLA-flax printed at various printing temperatures keeping constant the bed temperature.
Also in the discussion section we showed quantification of the main parameter effects approximated as linear effects for more clarity. Indeed, we needed a clear vision of the parameter influence which may not be captured through complex nonlinear approximation, we limited the cross-linear effects to first terms neglecting the nonlinear terms
Y=ax1+bx2+cx1x2 is approximated to Y=ax1+bx2 or Y=A+Bx1/x2
Amendment in new section 4 “The main affects are analyzed in this section with respect to first linear terms to achieve more clarity about the main observed effects of PLA and PLA-flax feedstock materials. This means that nonlinear terms are neglected in this analysis.”
Finally, if there are replications, are the values portrayed in figures 3, 6, and 11 mean values? Furthermore, which deviations were observed by the authors?
The tensile responses in Figures 3, 6 are not averages. This is why we specified this in the former version “The typical tensile performance…”. These can’t be averages simply because the specimen response should demonstrate the main features that occur during loading such as mechanical instability or plasticity. Averaging these curves will result in smoothing genuine effects, which we wanted to avoid. However, the data reported in Figure 11 are averages based on 4 replicates.
The deviation is now added in the present version of the manuscript.
Amendment: “of one replicate” added to describe figure 3 and 6. And “Figure 11 shows the average measured”. + section 4: “The average values for each mechanical property are obtained from four replicates of the considered printing parameter combinations. The analysis of scatter measured as the average of the standard deviation over the average value shows the following values 4%, 5% 5%, 7%, and 5% for PLA Young’s modulus, yield stress, tensile strength, ultimate stress and elongation at break. This concludes on stable and reproducible results for PLA. The same is, in fact, achieved for PLA-flax where the scatter values are slightly higher: 1%, 3%, 3%, 14%, 7%.”
- Page 3, line 93. Why have the authors chosen printing temperatures "slightly above" the manufacturer's recommendations? And Why those specific values?
We selected this values because there is a different range for PLA and PLA flax. As the reviewer may notice the range for PLA flax is significantly larger than for PLA. Using the same range for PLA would result is unwanted effects described in more details in a former paper by the authors. The compromise found was to adjust the range of printing temperature to be able to compare both materials.
Amendment in section 2 : “These values represent a compromise because PLA-flax can be printed at higher temperatures compared to PLA. Using the same temperature range of PLA-flax for PLA is not recommended as a significant amount of stringing may occur.”
- Page 3, Table. It is Table 2. Layer height is included two times in the table. Finally, a graphical scheme about the filament deposition angle and the building sequence could help identify those parameters with part orientation in the printer bed and the angle for the linear infill pattern used.
We agree with the reviewer regarding his comment. We deleted multiple occurrence of layer height and we introduced a sketch showing the printing angle.
Amendment: Figure 1c added
Figure 1.c
- Section 3. Regarding the regression equations, did the authors try to accomplish a regression for the three parameters: angle, Tp, and Tb, at the same time?
Yes we run an automated routine for linear / nonlinear regressions and for the three variables we ended up with nonlinerar complex equations with a large number of fitting parameters. When comparing the number of parameters needed, the number of experiments and levels used, we concluded that such equations are not robust enough so we previligied simple linear functions that represent the interaction between the studied parameters.
Amendment in section 4: “The correlation between the mechanical properties and the printing parameters is studied through fitting procedure. Automated fitting routine based on both linear and nonlinear regressions performed considering the three printing parameters (TB, TP, Q) showed that nonlinear complex equations with a large number of fitting parameters are top ranking ones. When comparing the number of parameters needed to the number of experiments available and the parameter levels used, it can be concluded that nonlinear regressions even representing the entire space of the parameters are not robust enough to provide a clear picture of the main printing parameter effects.”
- Section 4, lines 378-379. In the experiments, did the authors identify any mechanical benefit of using flax? If yes, this is not clear in the previous sections.
No, with regards to the properties of pure PLA, all mechanical properties are below those of PLA. The benefit would be towards thermal properties which will be the subject of another contribution by the authors.
Amendment in section 5: “However, Figure 11 shows that even with this optimal condition, PLA-flax does not bring a larger mechanical benefit compared to pure PLA since all mechanical properties are below those of PLA. The benefit would be towards thermal properties which are beyond the scope of the present study.”
- Figure 4. Could the authors be able to substitute the black background with the white color?
We tried to modify the background as the reviewer recommended but it shows less contrasted details compared to the original black one. See figure below.
- Figure 7.d, Figure 8. b Which regions are zoomed?
A black frame is added in figures 7c and for Figure 8b the micrograph was taken from another spot and does not point to Figure 8a.
- Figure 11. Figures labels should be (a) and (b).
Sorry for the mistake it is now corrected.
